# Investigation of Multiphase Flow in a Trifurcation Microchannel—A Benchmark Problem

**DOI:** 10.3390/mi13060974

**Published:** 2022-06-20

**Authors:** Eugen Chiriac, Marioara Avram, Corneliu Balan

**Affiliations:** 1Laboratory for Micro- and Nano- Fluidics- L10, National Institute for R&D in Microtechnologies—IMT Bucharest, 126A, Erou Iancu Nicolae Street, Ilfov, 077190 Voluntari, Romania; marioara.avram@imt.ro; 2REOROM Laboratory, Faculty of Power Engineering, University “Politehnica” of Bucharest, 313, Splaiul Independenței, Sector 6, 060042 Bucharest, Romania; corneliu.balan@upb.ro

**Keywords:** microfluidics, CFD, multiphase flow, interface

## Abstract

The evolution of an interface between two immiscible liquids in a three-branch symmetric microchannel is numerically and experimentally investigated. The main goals of the paper are to correlate the numeric data with the experimental results for the tested flow case and to assess the quality of the VOF procedure to trace the interface using the Fluent commercial code. The focus of the experiments was to characterize the dynamics of the oil–water interface formed in the vicinity of the bifurcation, at the entrance in the main microchannel of 400 microns width and 50 microns height. The oil core surrounded by water is visualized and micro-PIV measurements are performed in water. Experimental results qualitatively and quantitatively confirm the 3D numerical simulations. We propose the present investigated flow as a benchmark case for the study of the interface in a branching microchannel geometry.

## 1. Introduction

In multiphase flow, the validation and verification of the flow field, and the presence of multiple interfaces in the microchannel are topics of high importance in computational fluid dynamics. The applications of multiphase microfluidics are found in domains such as chemical synthesis [1], bioanalysis [2], 3D printing [3], and graphene field-effect transistors (GFET) integrated into microchannels [4] to create sensors with high sensitivity [5]. The advantage of linking immiscible fluids with microfluidics leads to important advancements in the domain of a 3D cell culture [6], where spheroids can be quantified and analyzed by a data-driven approach [7]. In the domain of chemical engineering, the separation of phases [8] plays an important role in the extraction of ions [9,10].

When the fluids are immiscible, there are several methods used to model the interface. Two main directions can be followed: surface methods and volume methods [11]. In general, in both methods, the one-fluid formulation is used for the Navier–Stokes equation where a source term is added to take into account the effects of the surface tension, along with a differential equation that has to be solved at the interface. For the surface methods or interface tracking methods, the interface is either marked with particles or represented by the grid, and it is advected by the fluid at the interface. For the volume methods, or interface capturing methods, the interface is reconstructed by solving a transport equation at the interface. There are certain volume methods: markers in fluid, volume of fluid, level-set method [12], and hybrid methods [13]. Another technique that can be used to solve the multiphase flow is the lattice Boltzmann color gradient method [14], where the macroscopic flow system is predicted by simulating the discrete fluid elements. Despite its applicability, the main drawback of the method is that numerical errors may arise when there is a high contrast between the material properties of the fluids used. In this work, we focused on the volume of fluid (VOF) method and its mechanisms are presented in [15].

In microfluidics, the VOF method has been used thoroughly to investigate the multiphase flow. Edirisinghe et al. [16] used the VOF method to simulate the formation of microbubbles in a T-junction microchannel under the influence of an electric field. It was observed that when the frequency of the alternative current was increased, the microbubble diameter decreased. In a double Y bifurcation, Zhang et al. [17] investigated the generation and dynamics of Janus droplets by using a shear-thinning fluid. When the capillary number of the dispersed phase is increasing, the number of Janus droplets generated in the shear-thinning fluid increases, while in the case of a Newtonian fluid, the flowing regime switches from dripping to jetting. In OpenFOAM, Roohi et al. [18] used the VOF method to investigate droplet formation regimes in the T-junction microchannel. They use the compressive interface capturing scheme for arbitrary meshes (CICSAM) method implemented in OpenFOAM for obtaining a sharper interface between the phases, a method that is less computationally expensive than piecewise linear interface calculation (PLIC). A new model for interface reconstruction was proposed and validated by Shams et al. [19] to reduce the spurious currents at the interface. The new model called contour level surface force (CLSF) is tested against the classic models’ continuum surface force (CSF) and sharp surface force (SSF), and it computes at each timestep, the face-based interfacial force. Kleijn et al. [20] used the VOF method to perform the benchmark numerical simulations of two-phase flows. They used a Laplacian smoother and an interface compression to sharpen the interface and reduce the spurious currents.

In this paper, we perform two 3D VOF numerical simulations by using two VOF formulations, namely explicit and implicit, on a benchmark geometry. The numerical simulations are subsequently qualitatively validated, by comparing flow visualizations with phase contours, and quantitatively validated, by comparing the velocity and vorticity distributions obtained from the simulations with the experimental data obtained from micro-particle image velocimetry (μPIV).

## 2. Numerical Method

### 2.1. Benchmark Definition—Geometry Description

To study the interface between immiscible fluids, a trifurcation microchannel is used, and is presented in Figure 1. The microchannel has the following characteristic dimensions: the central branch has a width of 100 μm, the lateral branches have a width of 129 μm, and the main channel has a width of 400 μm, with an angle between the central branch and the lateral branches of 30${}^{\circ }$. The microchannel has a height of 50 μm and a length, before and after the junction, of 2 cm.

One of the main targets of the present work was to define a benchmark microchannel geometry which might be used to study the dynamics of the interface in relation to different applications. We took into consideration the following: the study of the interface under symmetric and non-symmetric perturbations. This is the reason why we designed a three-entrance microchannel symmetric geometry. The non-symmetry will be induced to the interface using different flow rates or fluids on the lateral branches. The angle of 30${}^{\circ }$ is very common in many configurations, e.g., the branches in a respiratory system, and the Y-microchannel configuration used in mixing [21]. Finally, we decided to have the main width of 400 μm and an aspect ratio of 0.125. At this dimension and ratio, the influence of gravity can be neglected, and the flow dynamics are very close to the Hele–Shaw flow. We chose the ratio between the central inlet channel and the main channel 1:4 because one of our targets is to also to study the stability of a viscoelastic fluid interface and this extension/contraction flow is used in non-Newtonian studies.

The working fluids used in this study were sunflower oil and deionized water. The material properties of the working fluids, density, and viscosity are presented in Table 1 and they were experimentally determined using a mass per volume method and a standard oscillatory test. The interfacial tension between them was determined using the pendant drop method.

### 2.2. Governing Equations

The problem of multiphase flow is numerically tackled using the volume of fluid method. The one momentum equation is solved for both phases, the continuity equation is solved for the whole domain, and the transport equation for the volume fraction function, α, is solved at the interface. Considering that both fluids are Newtonian, incompressible, and immiscible, the equations take the following form:(1)$$\rho \left(\frac{\partial v}{\partial t}+\left(v\cdot \nabla \right)v\right)=\rho g-\nabla p+\eta \nabla^{2}v+F,$$
(2)divv=0,
(3)$$\frac{\partial \alpha }{\partial t}+\nabla \cdot \left(\alpha v\right)=0,$$
where ρ and η are the averaged material properties, v is the velocity vector, *g* is the gravitational force, *p* is the pressure, F is the volume force, and α is the VOF function that tracks the interface. The volume fraction is used to average the material properties as follows:(4)$$\rho =\rho_{water}\cdot \alpha +\rho_{oil}\left(1-\alpha \right),$$
(5)$$\eta =\eta_{water}\cdot \alpha +\eta_{oil}\left(1-\alpha \right).$$

F is a source term in Equation (1) which includes the effects of interfacial tension and interface curvature [22]. Its formulation is based on the continuum surface force model proposed by Brackbill et al. [23], and has the following expression in the Fluent code [24]:(6)$$F=\frac{2\rho }{\rho_{water}+\rho_{oil}}\sigma \kappa \nabla \alpha ,$$
where σ is the interfacial tension and κ is the curvature computed from the local gradients of the volume fraction at the interface: (7)$$\kappa =\nabla \cdot \frac{\nabla \alpha }{\left|\nabla \alpha \right|}.$$

### 2.3. Mesh, Boundary Conditions and Initial Conditions

In Figure 1, the region of interest of the geometry is presented, as well as the boundary conditions. At the inlets, we have a velocity inlet, while at the outlet, the relative pressure is set to 0 and the walls have a no-slip boundary condition. Two meshes were used in the VOF simulations and their characteristics are presented in Table 2. Compared to the first mesh, the second mesh is coarser, but it has more elements in the region of interest. For a high-quality mesh, both the minimum orthogonal quality and maximum aspect ratio must be close to 1; thus, as it can be seen in Table 2, mesh #1 is better than mesh #2.

The initial condition for these simulations is the value of the velocity at the inlets. For the side branches of the microchannel, we have $v_{1}=v_{3}=0.1$ m/s and for the central branch we have $v_{2}=0.006$ m/s. Additionally, to shorten the simulation time, the microchannel is filled with deionized water, while the second inlet is patched with sunflower oil.

Given that the VOF method is very numerically expensive, the grid test was performed using the same initial and boundary conditions described above, in laminar flow, with a single working fluid, water. The velocity distribution is plotted on two lines perpendicular to the flow field in Figure 2, one at the junction, and the other at 500 μm away, where the flow was fully developed. The results from the third simulation, where a coarser mesh was used, differ from the first two, which were performed on the finer meshes. As such, we proceed further using the first two meshes.

### 2.4. Numerical Details

To investigate the benchmark problem, the VOF method was used in the ANSYS Fluent numerical code. Two VOF formulations were tested with the same initial and boundary conditions. Both formulations can be used to solve transient flows. In both numerical simulations, the laminar flow model was used, since the Reynolds number for the aqueous phase is $Re_{w}=7$, while for the oil phase, it is $Re_{o}=0.007$.

The explicit formulation of the VOF method is specialized for transient flows with a constraint on the timestep and it was tested on mesh #1. This formulation is non-iterative and time-dependent, it has better numerical accuracy than the implicit formulation, but it has a limited timestep ($\Delta t=10^{-5}$ s for this simulation). The VOF method in this formulation has its Courant number set to 0.25. The interface modeling is sharp, and water is set as the primary phase, while sunflower oil is set as the second phase, because water has a higher density. The pressure–velocity coupling is performed using PISO, the transient formulation is first-order implicit and for the spatial discretization the following schemes were used: Green–Gauss cell-based for gradient, PRESTO! for pressure, QUICK for momentum and modified HRIC (high-resolution interface capturing) for the volume fraction.

The implicit formulation of the VOF method can be used for both steady or transient flows, which allows larger time steps and it was tested on mesh #2. The formulation is iterative, and the solution is achieved faster than in the explicit formulation (maximum allowed timestep for our simulation was $2\times 10^{-5}$ s). The interface is modeled in the same manner as in the explicit formulation. The first difference between the numerical setups comes from the pressure–velocity coupling. In the implicit formulation, the SIMPLE algorithm was used instead of PISO due to the fact that the PISO was diverging from the start of the simulation and the transient formulation was bounded second-order implicit. The spatial discretization is done using the following schemes: least squares cell-based for gradient, PRESTO! for pressure, second-order upwind for momentum and modified HRIC for the volume fraction. The second difference comes from the spatial discretization, where for the gradient method, a more expensive scheme compensates for the reduced number of elements of the second grid.

For the unsteady incompressible flows, the numerical solutions of the momentum and continuity equations are obtained using the velocity–pressure coupling methods SIMPLE and/or PISO. Following these procedures, the pressure term from the equation of motion is solved using a discrete Poisson equation; an iterative scheme is used for each time step until the divergence-free velocity is obtained with the imposed accuracy [25].

In both simulations, an adaptive timestep was imposed under the constriction that the Global CFL ≤1. The initial timestep size was $\Delta t=10^{-4}$ s, but after a few iterations, it was reduced to $10^{-5}$ s for the explicit simulation, and $2\times 10^{-5}$ s for the implicit simulation. Regarding the convergence, the desired numerical time was t=1 s. The residuals were set to $10^{-8}$, and after 1 s, the mass flow rate balance was checked between the inlet and the outlet. This resulted in the net mass flow rate of $-2\times 10^{-9}$ kg/s.

### 2.5. Results

In Figure 3a, the contours of velocity magnitude are presented. The contours are displayed on six planes that are perpendicular to the flow. The distance between the planes is 100 μm. It is notable from the first planes that the velocity profile is not yet fully developed. At the final planes, placed at 400 and 500 μm away from the junctions, the contours look similar. The maximum velocity recorded in the case of the trifurcation microchannel is 0.24 m/s. In Figure 3b, the contours of vorticity magnitude are presented. The color map was capped at 20,000 1/s to see where the maximum values of vorticity appear.

As expected, the maximum values of vorticity are at the walls. The maximum values of vorticity for the aqueous phase are recorded at the walls which are parallel to the flow field; this might be due to the low aspect ratio of the microchannel. We chose to enter with a more viscous fluid in a less viscous medium $\eta_{oil}/\eta_{water}=55$ in order to avoid the occurrence of the Saffman–Taylor instability [26].

In Figure 4, the velocity vectors are presented starting from a plane of 500 μm away from the junction. The velocity vectors are presented at a doubled scale to show the parabolic profile of the aqueous phase and its influence on the oil phase. A phenomenon was observed at the interface, due to the high difference in the magnitude of the velocity between the two phases, the interface acts as a moving wall and it drags the oil phase, thus resulting in an accelerated flow near the interface for the oil phase.

We begin the comparison between the two numerical simulations, where S1 is the explicit VOF simulation performed on the 2.5 M elements mesh while S2 is the implicit VOF simulation performed on the 1M elements mesh. In Figure 5, the phase contour is presented alongside the grid spacing for each simulation, and it can be seen that the interface in S1 is more diffuse than the interface in S2, given the clear difference between the two meshes. Furthermore, a diffuse interface is related to the spurious currents generated at the interface [27], and for the explicit VOF, more spurious currents were generated. The S2 mesh has more elements after the junction than S1 mesh.

In Figure 6, the interface is reconstructed and the lines of constant volume fraction are presented for both simulations starting from the junction (x=0μm) to x=500μm on the x axis. The previously reported numerical diffusion can now be observed for the S1 simulation, where the interface band has a thickness of 19.31 μm, compared to 12.77 μm, the thickness of the S2 band interface. However, when the interface lines of constant volume fraction, α=0.5, are compared, the two distributions of interfaces are in very good agreement.

In Figure 7, the distributions of velocity and vorticity magnitude are presented. The evolution of the velocity distribution from the entrance of the flow in the main channel to 500 μm away from the junction is presented. At the 500 μm mark in the channel, the flow is fully developed, and it has a parabolic distribution for the aqueous phase.

Concerning the vorticity distribution, we wanted to showcase the spikes in vorticity at the interface between the two fluids, and spikes that are comparable with the values of the vorticity magnitude at the wall, when the flow is fully developed. For the velocity distributions, the two numerical simulations match very well. However, a small difference is reported for the vorticity distributions, and the vorticity has a higher magnitude in the S2 simulation both near the wall and at the interface.

In Figure 8, the static pressure variation on the interface lines of constant volume fraction is presented. The two distributions have the same aspect, with the same pressure variation in the vicinity of the junction on all three lines, and the same pressure drop, from x=0μm to x=500μm, ΔP≈500 Pa. The interface is band, from α=0.1 to α=0.9, and the values in static pressure match on all three volume fractions after 100 μm; therefore, the variations in static pressure recorded in the vicinity of the junction are mainly induced by the curvature of the interface, in accordance with the Laplace equation. We also remark that the influence of the contact angle wetting might be important since the contact/wetting angle has influence over the local interface curvature, as it will be shown in the following section. However, the main difference between the two numerical simulations comes in terms of the magnitude of the static pressure, as the values obtained from the S1 simulation are ≈two times higher than the values obtained from the S2 simulation. The difference in static pressure does not come from the mesh, but from the different algorithms used to couple the velocity with pressure. In the explicit simulation, the PISO algorithm was used, an algorithm that is recommended for transient flows. In the implicit simulation, the SIMPLE algorithm was used, since for PISO, the implicit simulation was diverging very rapidly.

Then, on the same three-interface lines of constant volume fraction, the shear stress is analyzed. The shear stress is defined as:(8)τ=η·γ,
where γ is the strain rate. In Figure 9, the variation of the shear stress is presented. In both simulations, the shear stress recorded on the interface line, when α=0.5, has the highest variations. There is a slight difference between the two numerical simulations, in terms of the magnitude of the shear stress. In the vicinity of the aqueous phase, when α=0.1, the values obtained at the interface for shear stress are higher than the values obtained in the vicinity of the oil phase, when α=0.9. This can be explained by the amount of space that each phase occupies in the channel. The oil phase has a Reynolds number equal to 0.007, when the transition is made from the central channel to the main channel, the flowing surface expands and the velocity decreases; as such, the velocity gradients are smaller. Opposed to that is the aqueous phase which enters from the side channel into the main channel with a Reynolds number equal to 7. The flowing surface is shrunk, the velocity increases, and the gradients recorded are higher.

Given that the highest variations in static pressure and shear stress were recorded in the proximity of the junction, in the first 20 μm, they are further analyzed on five perpendicular lines to the flow field, at h=25μm for static pressure, and at h=0μm for the shear stress (on the top wall).

In Figure 10, the static pressure distributions are compared. As stated before, there is a difference between the two numerical simulations, the magnitude of the static pressure from S1 is almost two times larger than the magnitude of static pressure from S2. However, in both distributions, the pressure jump is recorded at the junction, meaning that the interface has a curvature. Approximating the oil phase with a cylinder, the principal radii of curvature can be determined from Laplace pressure:(9)$$\Delta P=2\kappa \sigma =\frac{\sigma }{R_{1}},$$
where the curvature $\kappa =1/R_{1}+1/R_{2}$ is reduced to $1/R_{1}$, at some distance from the junction, with $R_{2}$ going towards infinity after some distance from the junction. We must remark that, in the near vicinity of the junction, the two radii of curvature have finite values, which determined the pressure variations represented in Figure 8. From the numerical simulations, we obtain the following pressure differences: for S1 $\Delta P_{S1}=121.7$ Pa and S2 $\Delta P_{S2}=151.72$ Pa, the curvature radii obtained from the numerical simulations are $R_{1S1}=205.4\;\mu m$ and $R_{1S2}=164.8\;\mu m$.

In Figure 11, the distribution of the shear stress is presented on five lines on the top wall, h=0μm, from the junction with step of 5 μm to 20 μm away from the junction. In both simulations, the spikes of shear stress are recorded in the presence of the interface, and since the sunflower oil is more viscous than water, the values of shear stress are higher in the oil phase. There are differences between the two numerical simulations, and in the areas where the extreme values occur, higher values are recorded in the S2 simulation.

In Figure 12, the shear stress distribution is presented on the sidewall of the main channel at h=25μm, starting from the junction to 500 μm. In both simulations, the highest values of shear stress are recorded at the junction. The sudden drop in magnitude after the junction can be explained by the presence of the curved interface. At the 500 μm mark, the interface is straight, the pressure in the oil phase is equal to the pressure from the aqueous phase, and the shear stress distributions tend towards a constant value.

## 3. Microfabrication and Experimental Validation of the Benchmark Problem

### 3.1. Microfabrication

The microchannel is microfabricated by using PDMS (polydimethylsiloxane) and soft lithography. The microchannel mold is created in two steps: the first step is to display and develop a positive photoresist on a silicon wafer followed by the exposure of the photolithographic mask, with the microchannel design, on the silicon wafer by UV light; the second step is to etch the wafer using deep reactive ion etching (DRIE) with a Bosch process. The silicon surface of the mold is turned hydrophobic using 1 mL of chlorotrimethylsilane in a closed environment. The base and curing agent of PDMS are prepared in a 10:1 ratio. The mixture is degassed and poured on the microchannel mold. The ensemble is placed in the oven for an hour at a temperature of 90 ${}^{\circ }$C, as shown in Figure 13 left. The PDMS is peeled off the mold and drilled in the location of the microfluidic ports. The PDMS and a glass slide are placed in RIE (reactive ion etching) for an $O_{2}$ plasma treatment at a low power of 20 W for 20 s. The PDMS is sealed using the glass slide and the microfluidic ports are attached using epoxy, as presented in the center of Figure 13.

The obtained difference between the design and the microchannel is caused by the photolithographic step from the microfabrication process of the mold. In the photolithographic step, after the positive photoresist was developed, the only photoresist that remained on the silicon wafer was the one on the microchannel geometry. The recipe for the Bosch DRIE etching is standard; therefore, the reason for having a shrunk mold is that the photoresist is overdeveloped. This, in turn, is caused by having a geometry quite large for a microfabrication process. To obtain a perfect mold, either the geometry should have been slightly enlarged or the photolithographic step should have been optimized. The region of interest of the microfabricated PDMS microchannel is displayed in Figure 13 right.

### 3.2. Experimental Validation of the Benchmark Problem

To study the interface between immiscible liquids, two experimental setups were employed, one with fast cameras and one with µPIV. The first experimental setup consists of a syringe pump Harvard 33, CCD Camera and Photron Fast Camera, inverted microscope, and computer, while the second experimental setup has a µPIV acquisition system.

Using the first experimental setup, we qualitatively validated the numerical simulations with flow visualizations in Figure 14. The transition from velocity to flow rate is performed using the continuity equation. A precursor step is necessary before acquiring the image: instead of water, we introduce isopropanol into the microchannel, at the same flow rate, for cleansing the side branches of the microchannel and for dewetting the oil from the sidewalls. After a few minutes, the syringe of isopropanol was exchanged with one of pure water and the flow was let to stabilize. This extra experimental step is necessary since the velocities of the fluids involved are low and any impurity in the channel can deviate the interfaces.

Furthermore, the experimentally obtained image of the interface was digitized using ImageJ and the position of the interface is compared in Figure 15. The difference at the entrance in the main channel between the numerical interfaces and the experimental interfaces comes from the fact that the sunflower oil wets the wall at the junction. As such, the water phase cannot dewet the wall, and alcohol is needed, such as isopropyl alcohol to clean the PDMS wall. The reason why the interface is curved in the main channel is that the flow rates are low, and there is a difference between the contact angle of water with PDMS (≈$113^{\circ }$) and sunflower oil with PDMS (≈$45^{\circ }$).

For the micro-PIV experiments, the water phase is seeded in a volume concentration of 1% with fluorescent polyester particles that have a diameter $d_{p}=1\;\mu m$. Einstein’s relation for viscosity, when solid particles are introduced into a fluid is the following:(10)$$\eta_{m}=\eta_{0}\left(1+2.5V_{p}\right)=0.001\left(1+2.5\times 0.1\right)=0.00125\;\left[\mathrm{Pa}\cdot s\right],$$
where $\eta_{m}$ is the dynamic viscosity of water with the particles solution, $\eta_{0}$ is the viscosity of water and $V_{p}$ is the volume of particles introduced in water in cm${}^{3}$. The Einstein–Stokes equation for the low Reynolds numbers is used to compute the diffusion coefficient:(11)$$D_{b}=\frac{kT}{6\pi \eta_{m}d_{p}}=1.75\times 10^{-13}\;m^{2}/s,$$
where *k* is the Boltzmann constant and *T* is the temperature. The diffusion coefficient is used to assess the influence of the Brownian motion on the flow. The last parameter comes from the µPIV system and is the time between two laser pulses; for our experiment, Δt was found to be 40 μs. Using the average velocity of the aqueous phase U=0.1m/s, the relation used for computing the relative error related to the Brownian motion was proposed by Santiago et al. [28]:(12)$$\varepsilon =\frac{1}{U}\sqrt{\frac{2D_{b}}{\Delta t}}=9.35\times 10^{-4}.$$

Given that the value of the error is much lower than one, we neglect the influence of the Brownian motion.

In Figure 16, the micro-PIV process is displayed. A mask is applied on the set of images acquired, on the oil phase as well as on the exterior of the geometry; the background noise is removed by subtracting the mean image. The interrogation area chosen for this phenomenon was 32 × 32 pixels as it is important to have at least 10 particles in the interrogation box from one frame to another. The Average Correlation method is applied to the filtered set of images, and thus, a vector map is obtained.

The quantitative validation of the numerical simulation is presented in Figure 17 and it is performed by comparing the velocity distribution and the z-component of vorticity on a line perpendicular to the flow field, placed 500 μm away from the junction. From the µPIV experiments, the velocity distribution obtained has a parabolic profile and matches the velocity obtained from the numerical simulation. The uncertainty analysis was performed using the particle disparity method proposed by Sciacchitano et al. [29].

The z-component of vorticity is $\omega_{z}=\partial V/\partial x-\partial U/\partial y$ (where *V* is the spanwise component and *U* is the streamwise component of the velocity vector v) and its distribution is displayed in Figure 17 right. Compared to the vorticity magnitude distribution, here at the interfaces, we have a maximum followed by a minimum. The data points obtained from the experiments match the distribution obtained from the numerical simulation and as well their magnitude.

## 4. Conclusions

In this work, we investigated the evolution of the interface between two Newtonian immiscible fluids in a trifurcation microchannel using numerical simulations and experimental investigations. The liquid samples used in our study are oil and water with a viscosity ratio of 55. The viscous sample enters in trifurcation filled with water at a Reynolds number much lower than 1, to keep the stability of the interface. The numerical simulations were performed on 3D domains using the VOF method and the experiments consisted of flow visualizations and micro-particle image velocimetry. Two VOF formulations, explicit and implicit, were tested against each other, and even though a more refined mesh was used for the explicit formulation, in the implicit formulation the interface was sharper. At the junction, the domain was more refined for the implicit VOF. There is another reason for the less diffusive interface and it is that in the implicit formulation, the solution is iterative, and as was mentioned in the literature [19], the iterative solutions generate a narrow interface bandwidth. Therefore, this implicit VOF method is indicated to be used in the modeling of laminar interfacial flows, especially in the case of very low Reynolds numbers. When compared on a perpendicular line to the flow field, 500 μm away from the junction, the velocity distribution has the same shape and magnitude in the two simulations. Differences appear in the pressure comparison where the values recorded for the explicit formulation are two times higher. The insight we obtain from the shear stress distributions is that the value is increasing in the presence of the interface. The experimental device is microfabricated using soft lithography and PDMS. The phase contours obtained from the numerical simulations are matched by the flow visualizations, with the exception of the very vicinity of the junction, where the influence of the wetting/contact angle is more relevant. The numerical simulations are quantitatively validated by: (i) the velocity distribution, the same parabolic profile being obtained in the aqueous phase using the micro-PIV experiments; and (ii) the experimental values of the z-component of the vorticity match the numerical distribution. The same peaks for vorticity that are numerically observed in the presence of the interface are experimentally obtained.

The goal of this study was to design, test, and propose a benchmark geometry for the study of liquid interfaces, both numerically and experimentally. The numerical results of the analyzed case were qualitatively and quantitatively confirmed by experiments. Using the VOF implicit scheme for computation has been proven to be less time-consuming. Further investigations of the dynamics of the interface are focused on the entrance of the viscous fluid into the junction for different contact angles and viscous fluids.

## Figures and Tables

**Figure 1 micromachines-13-00974-f001:**
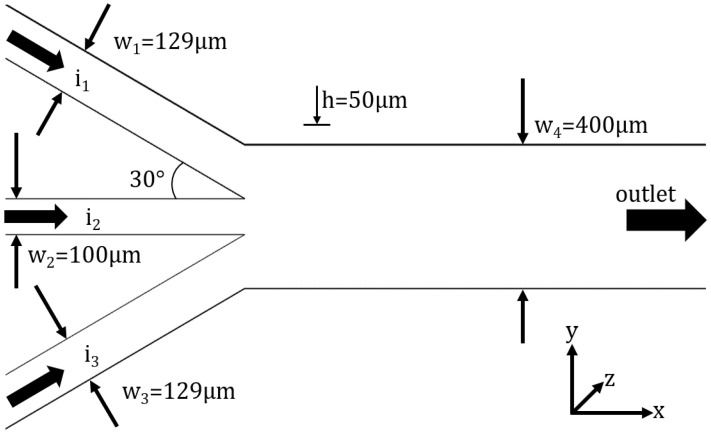
Microchannel geometry. The same fluid is introduced in branches 1 and 3 with the same flow rate, therefore we expect a symmetric interface in this configuration.

**Figure 2 micromachines-13-00974-f002:**
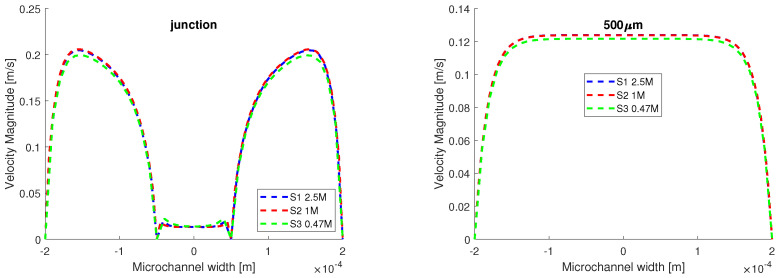
Grid test using homogeneous flow. The test fluid is water and the input velocities are: $v_{1}=v_{3}=0.1$ m/s and $v_{2}=0.006$ m/s. There are minor differences between the results obtained from S1 and S2.

**Figure 3 micromachines-13-00974-f003:**
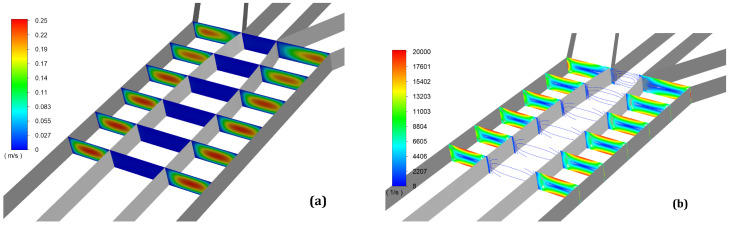
Velocity magnitude (**a**) and vorticity magnitude (**b**) contours on six perpendicular planes to the flow field with a distance between them of 100 μm. We can observe that the flow is stabilized after 300 μm from the junction, because we chose laminar flow with a high ratio between the Reynolds number $Re_{oil}/Re_{water}=10^{-3}$ for a stable interface.

**Figure 4 micromachines-13-00974-f004:**
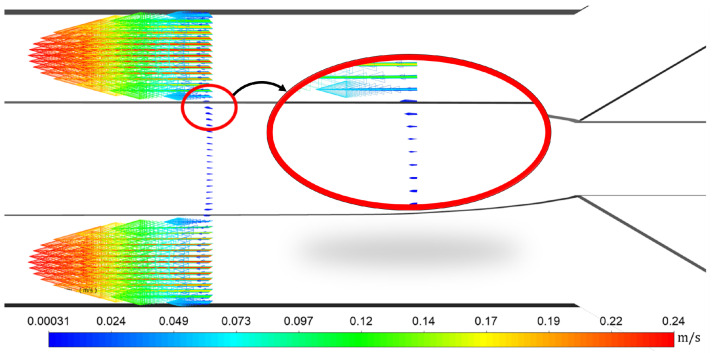
Velocity increasing in the oil phase in the vicinity of the interface.

**Figure 5 micromachines-13-00974-f005:**
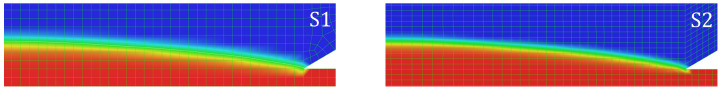
Details of the interface phase contours; the difference between the two simulations S1 and S2, respectively. The interface thickness (ϕ) is the difference between the volume fractions, α=0.1 and α=0.9, and for S1 $\phi_{S1}=19.3\;\mu m$, while for S2, $\phi_{S2}=12.8\;\mu m$.

**Figure 6 micromachines-13-00974-f006:**
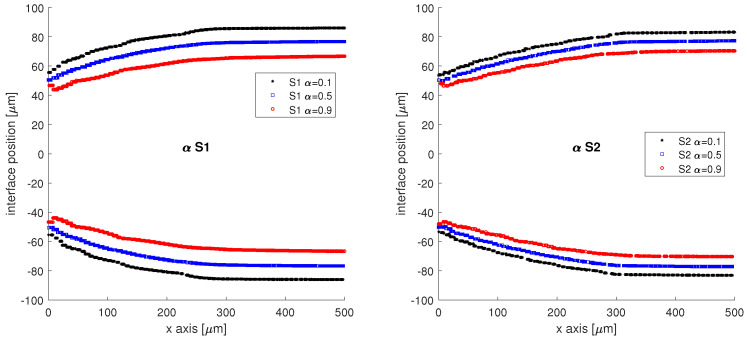
Interface reconstruction on the lines of constant volume fraction of the oil phase, starting from x=0μm to 500μm away from the junction; α=0.5 is considered to be the interface.

**Figure 7 micromachines-13-00974-f007:**
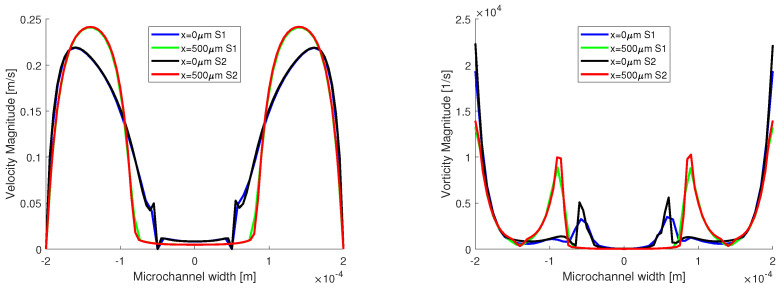
Velocity and vorticity distribution on two lines inside the channels.

**Figure 8 micromachines-13-00974-f008:**
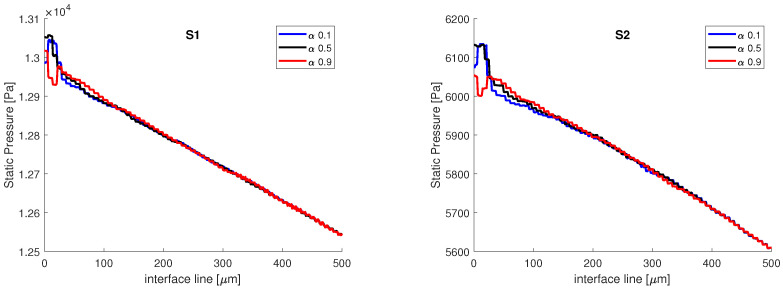
Static pressure variation on the interface lines at different volume fractions. The main differences are observed in the near vicinity of the junction.

**Figure 9 micromachines-13-00974-f009:**
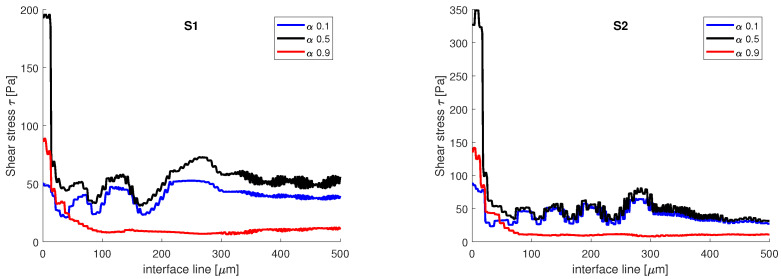
Shear stress variation on the interface line at different volume fractions.

**Figure 10 micromachines-13-00974-f010:**
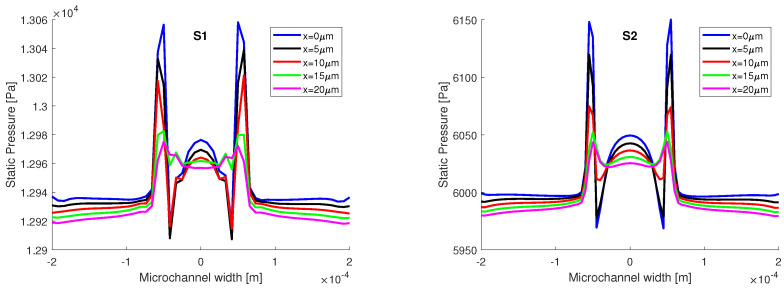
Distributions of static pressure on 5 lines in the middle plane at h=25μm.

**Figure 11 micromachines-13-00974-f011:**
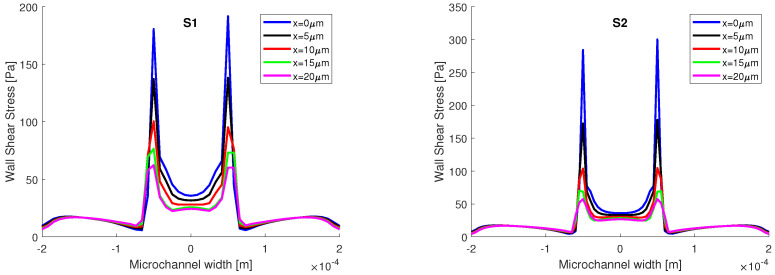
Wall shear stress distribution on 5 lines on the top wall at *h* = 0 μm.

**Figure 12 micromachines-13-00974-f012:**
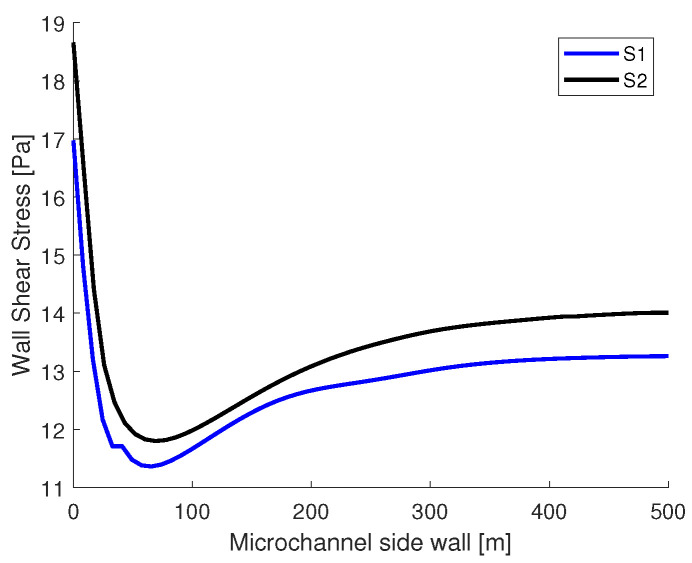
Wall shear stress distribution on the sidewall at *h* = 25 μm at the intersection with the middle plane.

**Figure 13 micromachines-13-00974-f013:**
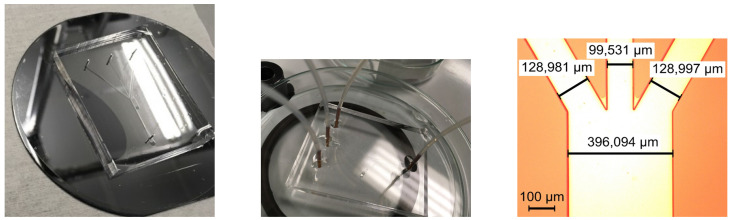
Microfabrication process: PDMS microchannel on silicon mold and final aspect of the microfluidic device and microscope detail on the region of interest.

**Figure 14 micromachines-13-00974-f014:**
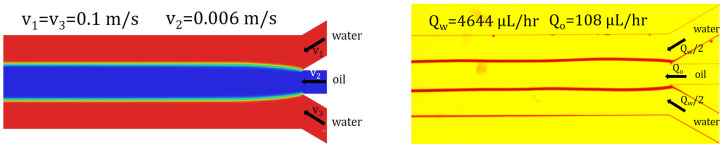
Experimental interfaces—qualitative validation of the numerical simulation using flow visualization. The input flow rates in the experiments correspond to the average velocities magnitude imposed in the numerical simulations.

**Figure 15 micromachines-13-00974-f015:**
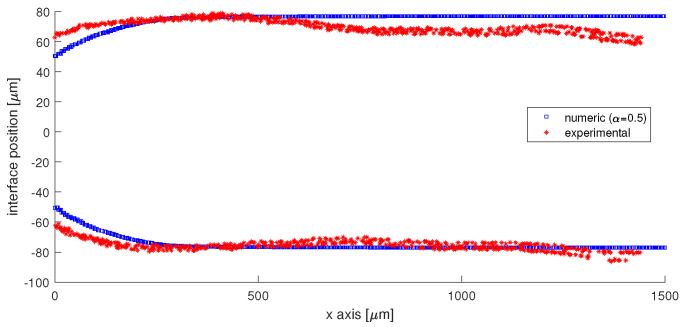
Quantitative validation of the position of the interface. The difference at the junction is generated by the different wetting angle of water and oil at the PDMS surface, in comparison with the contact angle of 90${}^{\circ }$ imposed in numerical simulations. A slight asymmetry of the interface is experimentally recorded.

**Figure 16 micromachines-13-00974-f016:**
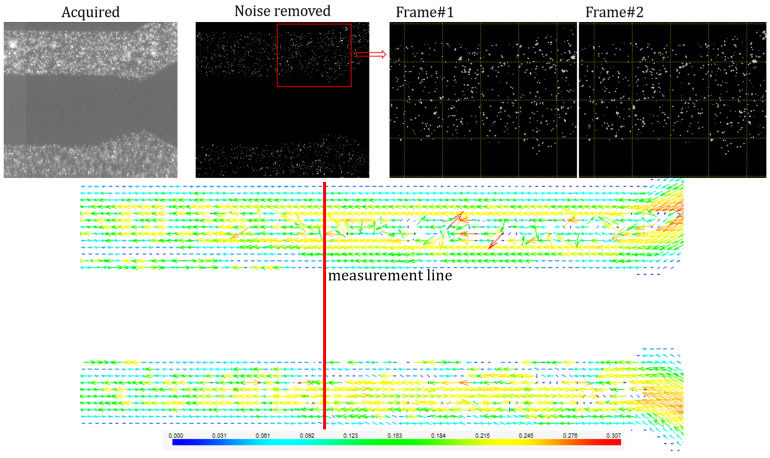
Micro-PIV measurement—image acquisition and processing, vector map in the aqueous phase.

**Figure 17 micromachines-13-00974-f017:**
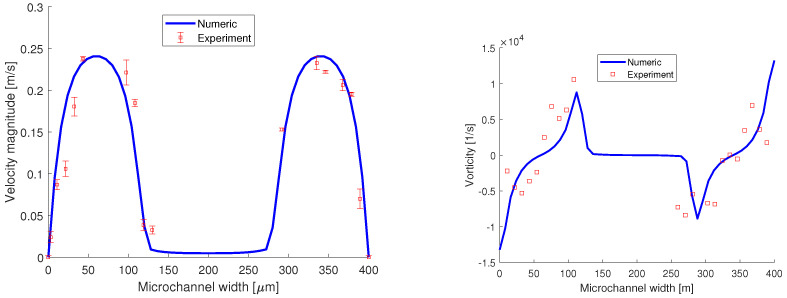
Quantitative validation of the numerical simulation using velocity and vorticity distributions, obtained from the micro-PIV measurements, as can be seen in Figure 16.

**Table 1 micromachines-13-00974-t001:** Material properties at 25 degrees.

Sample	$\rho \;(\mathrm{kg}/m^{3})$	η(mPa·s)	σ(mN·m)
deionized water	1000	1	25
sunflower oil	925	55	

**Table 2 micromachines-13-00974-t002:** Mesh properties.

Mesh	Cells	Faces	Nodes	Min. Orth. Quality	Max. Aspect Ratio
#1	2,508,930	7,695,921	2,680,942	0.62	2.9
#2	960,560	2,961,014	1,041,488	0.55	6.7
#3	470,700	1,476,480	536,822	0.81	6.7

## Data Availability

All experimental and numerical data are available upon reasonable request from the corresponding authors via email.

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
