# Peer review of "Investigation of Multiphase Flow in a Trifurcation Microchannel—A Benchmark Problem"

_micromachines, 2022, doi:10.3390/mi13060974_

Round 1

Reviewer 1 Report

Comments to the authors:

1.     Why different spatial schemes for gradient computation were used with different time schemes?

2.     Why is there a maximum timestep size for an implicit scheme?

3.     Why static pressures are different for two meshes?

4.     A grid convergence study should be done to verify that the simulation results are correct.

Author Response

Dear Reviewer,

Attached you will find our answers.

Best regards,

Reviewer 2 Report

Ms. Ref. No.: Micromachines-1757664

Title: Investigation of Multiphase flow in a Trifurcation Microchannel - A Benchmark Problem

In this paper, numerically and experimentally the interface evolution problem in a microchannel.  The numerical simulations are performed using the Volume of Fluid method and they are qualitatively and quantitatively validated. The experiments consist in flow visualizations and micro-Particle Image Velocimetry. The current test case is a benchmark problem in microfluidics, and it is represented by a  separated flow with two interfaces. The analyses themselves are sound and I believe the results of their work. The data are well organized by the authors. I, therefore, recommend this paper be published in the Micromachines Journal after the authors address the following comments.

·         Review English grammar as there are mistakes throughout the text. This article should be completely rewritten.

·         An abstract is not well organized. The abstract must be improved. The authors must explain the application and novelty of the research work add in the abstract section.

·         The literature section must be improved with more advanced articles and clearly why your present study is different, better to explain novelty.

·         More physical insight into the discussion section is needed.

·         The physical explanation of figures 10 -12 is limited. Please explain more

·         More physical insight into the discussion section is needed.

·         What is time used for convergence?

·         What is the criterion of convergence?

·         Where is mesh independency?!

·         The authors used the laminar flow or turbulence flow? Please explain in the text.

·         The literature section must be improved with more advanced articles and clearly why your present study is different, better to explain novelty.

·         The author must improve the introduction with more advanced applications. Also, the author could find new references for the literature review. For example:

Chemical Engineering Journal 356 (2019): 492-505.

Separation science and technology 54, no. 15 (2019): 2536-2554.

Analytica chimica acta 838 (2014): 64-75.

Chemical Engineering Journal 334 (2018): 2603-2615.

Chemical Engineering and Processing: Process Intensification 108 (2016): 35-43.

AIChE Journal 61, no. 6 (2015): 1912-1924.

Chemical Engineering Journal 328 (2017): 1075-1086.

Separation and Purification Technology 231 (2020): 115875.

In conclusion, this paper might be made suitable for publication in this Journal if the as-mentioned comments are clarified. These constitute a major revision of it.

Author Response

Dear reviewer,

Attach you will find our answers.

Best regards,

Reviewer 3 Report

This manuscript discussed multiphase flow in a trifurcation microchannel, authors studied this problem using simulations and verified with experimental results. This is an important problem to study, however, the authors investigated this question only under one simple condition; all the important parameters (e.g., geometric parameters and flow rate), which could significantly influence the flow profile, have not been discussed. Therefore, significant revisions are needed before considering for publication.

(1)   The authors need to justify the geometrical design of the system. For example, why w1=129nm, w2=100nm, and w3=129nm, and 30 degree were chosen to study? The ratio of three channel size and angle degree could significantly influence the flow development, so these parameters should be carefully investigated and discussed.

(2)   Similarly, in this manuscript, the authors only chose one velocity condition to study, v1=v3=0.1m/s, v2=0.006m/s. However, the relative flow rate between the oil phase and the water phase is the key parameter to influence the flow profile, especially at the fluid interface. Therefore, the influence of flow rate should be carefully investigated.

Author Response

Dear reviewer,

Attached you will find our answers,

Best regards,

Round 2

Reviewer 1 Report

1. Maybe the discussion on the pressure can be removed from the manuscript, since the pressure obtained from the semi-implicit pressure projection methods (SIMPLE and PISO) does not represent any physical pressure.

Author Response

Dear reviewer, we thank you for your comment. We are aware that computed pressure using SIMPLE or PISO schemes (considered in this case a free scalar for the momentum equation) is a quantity which assures the convergence of the incompressible velocity field, not directly related to the “physical pressure”. However, the computed pressure is important in our numerical study as a measure which makes the difference between the two VOF formulations.

Reviewer 2 Report

It is my personal opinion than the paper is ready for publicaction.

Author Response

We thank the reviewer for his valuable comments.

Reviewer 3 Report

The authors addressed the comments, I recommend the publication of this manuscript.

Author Response

(The authors gave the same response as above.)
